# Two Dimensions of Nutritional Value: Nutri-Score and NOVA

**DOI:** 10.3390/nu13082783

**Published:** 2021-08-13

**Authors:** Carmen Romero Ferreiro, David Lora Pablos, Agustín Gómez de la Cámara

**Affiliations:** 1Scientific Support Unit (i+12), Hospital Universitario 12 de Octubre, 28041 Madrid, Spain; david@h12o.es (D.L.P.); acamara@h12o.es (A.G.d.l.C.); 2Clinical Research Support Network ISCIII, 28040 Madrid, Spain; 3Faculty of Biological Sciences, Universidad Complutense de Madrid (UCM), 28040 Madrid, Spain; 4Consorcio de Investigación Biomédica en Red de Epidemiología y Salud Pública (CIBEResp), 28029 Madrid, Spain; 5Faculty of Statistics, Universidad Complutense de Madrid (UCM), 28040 Madrid, Spain; 6Faculty of Medicine, Universidad Complutense de Madrid (UCM), 28040 Madrid, Spain

**Keywords:** ultra-processed foods, Open Food Facts, food labelling, food quality, nutrition

## Abstract

Front-of-pack labels can improve the ability of consumers to identify which foods are healthier, making them a useful public health tool. Nutri-Score is a front-of-pack labelling system adopted by several European countries. This system ranks foods according to their nutritional quality, but does not consider other dimensions such as the degree of food processing. The aim of this study is to compare the nutritional quality (as assessed by Nutri-Score) and the ultra-processing (as assessed by the NOVA classification) of foods in the Open Food Facts database. A simple correspondence analysis was carried out to study the relationship between the two systems. Ultra-processed foods (NOVA 4) were found in all Nutri-Score categories, ranging from 26.08% in nutritional category A, 51.48% in category B, 59.09% in category C, 67.39% in category D to up to 83.69% in nutritional category E. Given the negative effect that the consumption of ultra-processed foods has on different aspects of health, front-of-pack labelling with Nutri-Score should at least be accompanied by complementary labelling indicating the level of processing, such as the NOVA classification.

## 1. Introduction

One of the main objectives of public health in developed countries is to prevent the growth of the most prevalent chronic non-communicable diseases (NCDs). NCDs now account for more than one-half of the global burden of disease. In recent decades, dietary habits have changed, and these changes have been paralleled to an increase in NCDs [1,2,3]. Those diseases are highly related to the excessive or unbalanced consumption of certain foods and/or nutrients [4,5,6]. Spain, despite following a Mediterranean diet with beneficial health effects [7,8], has to deal with nutritional problems that have a substantial human, social and economic cost.

To achieve an improvement in nutritional status and avoid chronic nutrition-related diseases, international organisations have recommended several strategies. Nutritional labelling is one of the tools proposed by public health policies to promote healthy dietary choices, which is regulated by the European Union. This is a cost-effective tool for communicating nutritional information to consumers at the point of sale, allowing them to better understand the food they buy and consume [9]. Although food labelling is mandatory in most countries, the application of nutrition labelling differs from country to country. In November 2018, the Spanish Ministry of Health announced the official adoption of the front-of-pack nutrition label Nutri-Score [10]. This type of front-of-pack labelling was also adopted in France and Belgium and is under discussion to be adopted by other European countries. The implementation of new labels is due to the fact that the current nutritional information on food packaging is difficult to read and understand for consumers, and most of them do not use this information during their purchases [11,12]. This new front labelling is a coloured logo associated with letters that describe nutritional quality. Nutri-Score divides foods into five categories ranging from A (most healthy) to E (least healthy) depending on their nutritional quality. This labelling system takes into account the nutrient composition as the excessive intake of some nutrients has negative effects on health [13,14,15,16,17,18]. Theoretically, Nutri-Score allows consumers at the time of purchase to easily assess the nutritional quality of foods and also encourages manufacturing companies to improve the nutritional composition of their food products [19].

In the last few years, some attention has been paid to the growing importance of food processing. NOVA [20,21] is a food classification system that divides foods according to the degree of processing, rather than in terms of nutrients. Processed foods, according to the NOVA classification, are products made from unprocessed or minimally processed foods with added oil, sugar, salt or other common culinary ingredients. The term “ultra-processed food” was introduced by NOVA. This term is intended to identify industrially formulated products made from substances extracted from food or synthesised in laboratories. Ultra-processed foods are usually products with a lower nutritional quality [22,23,24,25]. Recently, it has been shown that the consumption of ultra-processed foods leads to NCDs such as obesity, hypertension, dyslipidaemia or cancer [26,27,28,29,30]. In addition, some studies reported results on the negative effect of ultra-processed food consumption on all-cause mortality [31,32,33,34,35]. Based on this, the NOVA system, proposed by Monteiro et al. [20,21], is widely used worldwide as a method to classify foods according to their degree of processing and to predict the risk of developing NCDs [36,37]. As both the consumption of ultra-processed foods and the prevalence of NCDs continue to increase, there is a need to provide more detailed nutrition information on food labels. In this way, a label that addresses the degree of food processing could be useful for public health and complementary to existing labels.

The aim of this study is to compare the classification of foods on the Spanish market from the Open Food Facts database using the Nutri-Score and NOVA systems and detect ultra-processed products that are misclassified by the Nutri-Score system.

## 2. Materials and Methods

### 2.1. Open Food Facts Database

The Open Food Facts database was used to obtain the ranking of foods using the NOVA and Nutri-Score systems (https://world.openfoodfacts.org/, accessed on 24 February 2021). Open Food Facts is a collaborative project of food products traded worldwide. This database has an Open Database License (ODBL) and contains nutritional data (ingredients, allergens, nutritional properties and all the information that can be found on product labels) on hundreds of thousands of products. For this study, data were collected from the Open Food Facts database on 24 February 2021. Duplicates (or different presentations, for example, packs “×4” or “×6”) for products of the same composition and brand were eliminated. All products currently marketed in Spain with Nutri-Score and NOVA classification were included (*n* = 9931, see flow chart in Figure 1).

### 2.2. Nutrient Quality Classification System of Foods—Nutri-Score

The nutritional quality of the food was established using Nutri-Score [10]. Nutri-Score is a front-of-pack nutrition label that translates the nutritional quality of a product into a five-letter code (A, B, C, D and E), each letter corresponding to a different colour. The algorithm on which Nutri-Score is based is on a continuous and discrete scale ranging from +40 (least healthy) to −15 (most healthy). Energy, total sugar, saturated fat and sodium score negative points, while fruit and vegetables, nuts and legumes, protein and fibre score positive points. The total sum of the score is divided into five groups (group A includes scores between −15 and −1, group B includes scores between 0 and 2, group C includes scores between 3 and 10, group D includes scores between 11 and 18, and group E includes scores between 19 and 40).

### 2.3. Degree of Processing Classification System of Foods—NOVA

The degree of food processing was established using the NOVA classification [20,21]. NOVA establishes four groups: group 1 collects unprocessed or minimally processed foods; group 2 describes processed culinary ingredients; group 3 comprises processed foods; group 4 includes all ultra-processed foods. The differences between the NOVA 4 and NOVA 3 groups are based on the type and amount of components and additives included, and the use of whole grains or refined foods. Ultra-processed foods (NOVA 4) are considered industrial preparations made from food-derived or laboratory-synthesised substances (flavourings, colourants and other common additives) that usually contain little or no whole foods.

### 2.4. Statistical Analysis

All statistical analyses were performed with SAS© software (SAS Institute Inc., Cary, NC, USA), Version 9.4 of the SAS System for Windows. Descriptive data were expressed as absolute or relative frequencies for categorical variables and continuous variables were presented as median and interquartile range (IQR). The relationship between the two categorical variables that classify foods (NOVA and Nutri-Score) was described by simple correspondence analysis. This relationship was represented by a ternary diagram. The ternary diagram is a triangular graph that visualises in a two-dimensional way the relationships between Nutri-Score (represented by dots in the diagram) and the percentage of ultra-processed foods according to the NOVA classification (represented on each of the three axes).

## 3. Results

Of the 9931 foods included in the study, according to Nutri-Score, the most frequent category was D with 25.96% followed by C with 23.03% of the total. According to the NOVA classification, the group with the highest frequency was ultra-processed foods (NOVA 4) with 56.45% (Figure 2). Of all foods classified as NOVA 4 in the database, 75.50% were classified as medium–low nutritional quality (C, D and E) by Nutri-Score.

All Nutri-Score groups included a considerable percentage of ultra-processed foods (NOVA 4), ranging from 26.08% in nutritional category A (the category of highest nutritional quality) to 83.69% in nutritional category E (the category of lowest nutritional quality) (Table 1).

In the ternary diagram (Figure 3), the NOVA 2 group is not shown because it only accounted for 1% of the total data. The axes of the diagram correspond to the percentage of foods belonging to NOVA 1, NOVA 3 and NOVA 4 (these percentages are also shown in Table 1). The dots represented inside the triangle correspond to the five Nutri-Score rankings (A, B, C, D and E) according to the amount of products they included from each of the different NOVA groups. As an example of interpretation, using the Nutri-Score category A represented with dashed lines in Figure 3, 40.34% of the foods classified in this category belong to unprocessed or minimally processed foods (NOVA 1), 33.52% correspond to processed foods (NOVA 3) and 26.08% to ultra-processed foods (NOVA 4). This interpretation can be performed in the same way for the rest of the dots in the diagram. As can be seen in the diagram, Nutri-Score categories B, C and D have a similar percentage of NOVA 3 and NOVA 4. In addition, NOVA 1 values are related in categories B, C, D and E.

As shown in the ternary diagram, even the Nutri-Score A category (highest nutritional quality category) included ultra-processed foods (26.08%). The main ultra-processed food groups included in the Nutri-Score A classification were: dairy products (21.92%), ready meals and canned dishes (18.72%), vegetarian/vegan ready meals (14.63%), flavoured plant-based drinks (10.68%) and pastries and cookies (8.06%) (Table 2).

When studying the Nutri-Score as a continuous score, as can be seen in Figure 4, the score underpinning Nutri-Score (the lower the score, the better the nutritional quality and, conversely, the higher the score, the lower the nutritional quality) is lower for unprocessed or minimally processed foods (NOVA 1), while for ultra-processed foods (NOVA 4) it is higher. Even so, the median values for the ultra-processed food group are quite low (11 (3 to 17)) on a scale where the upper limit is 40 points, and the categories indicating worse nutritional quality (D and E) range from 11 points to 40.

## 4. Discussion

Ultra-processed foods are products of lower nutritional quality [22,23,24,25], so most of these products should be classified in Nutri-Score as class C, D or E. In this study, we found that only 75.50% of ultra-processed foods (NOVA 4) are rated as medium–low nutritional quality (C, D and E) in Nutri-Score. These data are in line with those provided by another study where the authors found that 79.00% of ultra-processed foods were classified as medium–low nutritional quality by Nutri-Score [38]. It is noteworthy that all Nutri-Score categories include ultra-processed foods, from 26.08% in the highest nutritional quality category (A) to more than 50% in the rest of the categories from B onwards. Although the proportion of ultra-processed foods is in ascending order in the different Nutri-Score categories, this is quite shocking, as in the Nutri-Score B, C, D and E classifications, at least one out of two foods are ultra-processed (NOVA 4). Consequently, the Nutri-Score system fails to identify all unhealthy foods that can be rated well nutritionally, although they are highly processed and contain many additives. Several studies [39,40] found clear evidence that front-of-pack labels can improve the ability of consumers to identify which foods are healthier, making them a useful tool for public health. However, for these systems to be effective, labels must provide clear and complete information. The Nutri-Score front-of-pack label has several contradictions, since it does not cover all the health dimensions of the food, as we have seen in this study with the degree of food processing. No logo can include all dimensions in a single indicator. Nutri-Score only reports to consumers the overall nutritional quality of foods based on the nutrients they contain. This labelling system allows food to be compared without taking into account the other dimensions of food health, which prevents unhealthy foods from being correctly differentiated. Focusing on the degree of food processing, two products with the same Nutri-Score class can be either an ultra-processed food, or an unprocessed or minimally processed food according to the NOVA classification. If all the necessary information is not available, the consumer may interpret that both foods are equally good because their nutritional quality through Nutri-Score is the same. For example, we find in the Open Food Facts database that a natural yoghurt with just whole milk and active cultures as ingredients is classified as category A or B in the Nutri-Score. On the other hand, most fruit-flavoured yoghurts (e.g., strawberry yoghurts) with added sugar, flavourings and colourings are also classified as category A or B in the Nutri-Score. This leads to an interchangeable choice between the two types of yogurts, whereas the plain yoghurt is an unprocessed or minimally processed food, and the fruit-flavoured yoghurt is an ultra-processed yoghurt with sugar and other additives. The first one is clearly a healthier choice. The same applies to other products such as legume ready meals of different brands, which are high in additives and ultra-processed foods but are classified as category A or B in Nutri-Score, as well as natural pulses or canned pulses, which are unprocessed or minimally processed foods.

It should be noted that the Nutri-Score A category contained a limited but significant number of unhealthy foods (more than a quarter of the foods labelled as A). The main ultra-processed foods included in this category were dairy products or ready meals containing added sugars and additives [41,42]. In addition, most of these foods were plant-based, which is generally considered healthy by consumers [43], so special care should be taken when selecting this type of food. 

Food processing has an impact on the food matrix and affects the functionality of foods [44], making them unhealthy foods despite satisfying the nutritional properties estimated from their nutrient content. This can be seen in the data obtained in Figure 4, where foods classified as NOVA 4 had relatively low Nutri-Score values (11 [3,4,5,6,7,8,9,10,11,12,13,14,15,16,17]). These foods may contain nutrients that are rated positively, but the Nutri-Score does not take into account the level of processing they have undergone and the added additives that make them unhealthy. The weight of processing seems to be even greater than the nutrient content, as we eat complex food matrices and not nutrients. Thus, enteral and parenteral nutrient solutions, even if balanced with all known nutrients, cannot compensate for the physiological effect of the food.

Traditionally, the Spanish lifestyle has been based on the Mediterranean diet [45]. However, it is also known that in recent years the Spanish population has been moving away from the traditional dietary pattern towards a less healthy diet [46], which has led to nutritional problems that have an economic and social impact. In this study, we observed that more than a third of the foods on the Spanish market that we studied (3435 out of 9931) correspond to unhealthy foods (foods classified as D and E, and NOVA 4). The increase in the Spanish market of unhealthy foods (generally with aggressive advertising), in addition to the nutritional transition from local and traditional foods to industrial and globalised foods that Europe is experiencing, may be possible causes of these current nutritional problems. Knowing the existing evidence of the negative effect that the consumption of ultra-processed foods has on the different aspects of health [26,36,47] and the WHO recommendations to reduce the consumption of this type of food as much as possible, the front-of-pack labelling with Nutri-Score should at least be accompanied by other complementary labelling indicating the level of processing. Several tools are now available that focus on food processing [48], such as the SIGA classification [44,49], which classifies foods based on the NOVA classification and degree of processing, in addition to other factors. The application of these classifications in a label would allow consumers to know that they are choosing an ultra-processed food, and once the consumer knows this information, they could interpret the nutritional quality of the product through the Nutri-Score. This information not only affects ultra-processed foods, but it could also improve the choice of unprocessed foods. 

This study includes information on 9931 products present in the Spanish market from the Open Food Facts (2021) database. It includes foods of all types, not just generic foods, so we consider it to be a representative sample of the variability and quantity of the food supply in the stores of Mediterranean countries such as Spain.

## 5. Conclusions

Food processing and the nutritional quality of food cover different but complementary dimensions. All Nutri-Score categories include ultra-processed foods (at least 26% ultra-processed foods). Therefore, the information provided by Nutri-Score is incomplete and fails to identify all unhealthy foods. For this reason, labels that indicate the degree of food processing by applying the SIGA or NOVA classification should complement it to enable consumers to make better choices. Furthermore, any front-of-pack food labelling must be accompanied by an educational campaign aimed to raise awareness and guide the consumer to make healthier food choices.

## Figures and Tables

**Figure 1 nutrients-13-02783-f001:**
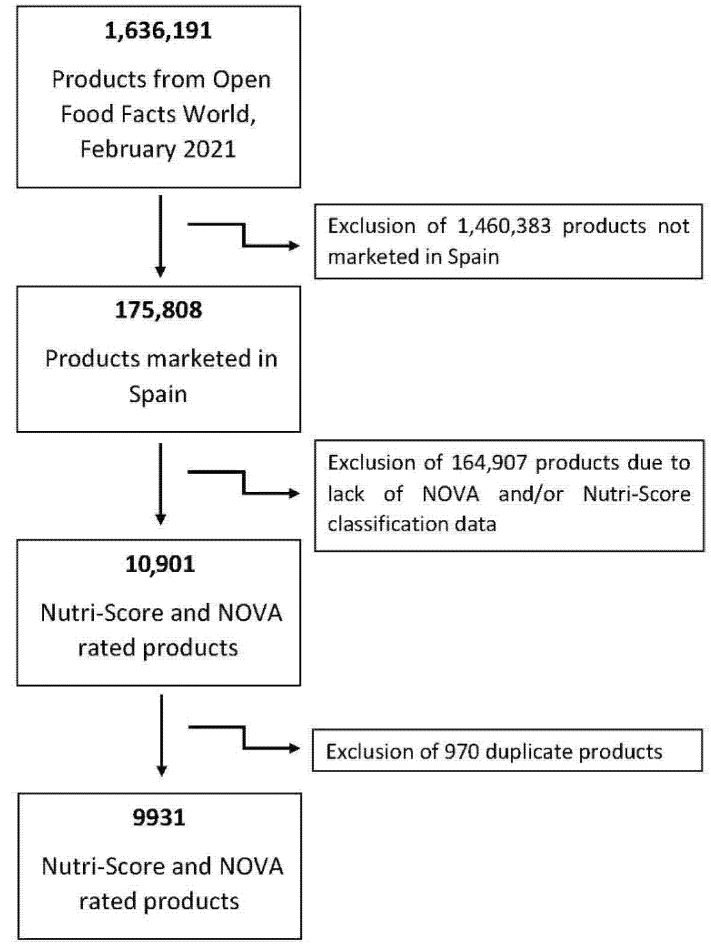
Flowchart of the total amount of foods included in the study.

**Figure 2 nutrients-13-02783-f002:**
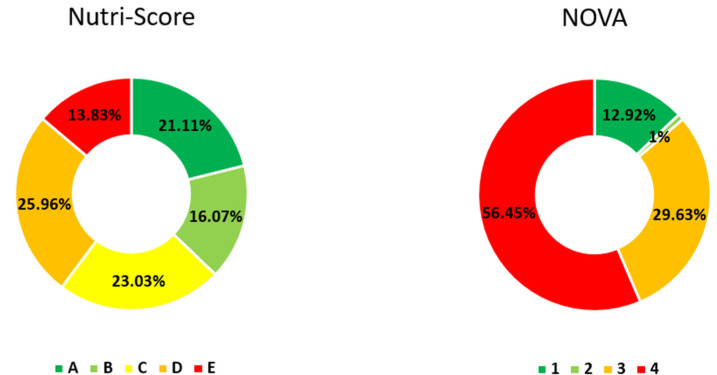
Distribution of foods according to Nutri-Score and NOVA classification groups. Nutri-Score groups (A, B, C, D, E). NOVA groups (1, 2, 3, 4). Numbers represent the percentage of total food included in the study (*n* = 9931).

**Figure 3 nutrients-13-02783-f003:**
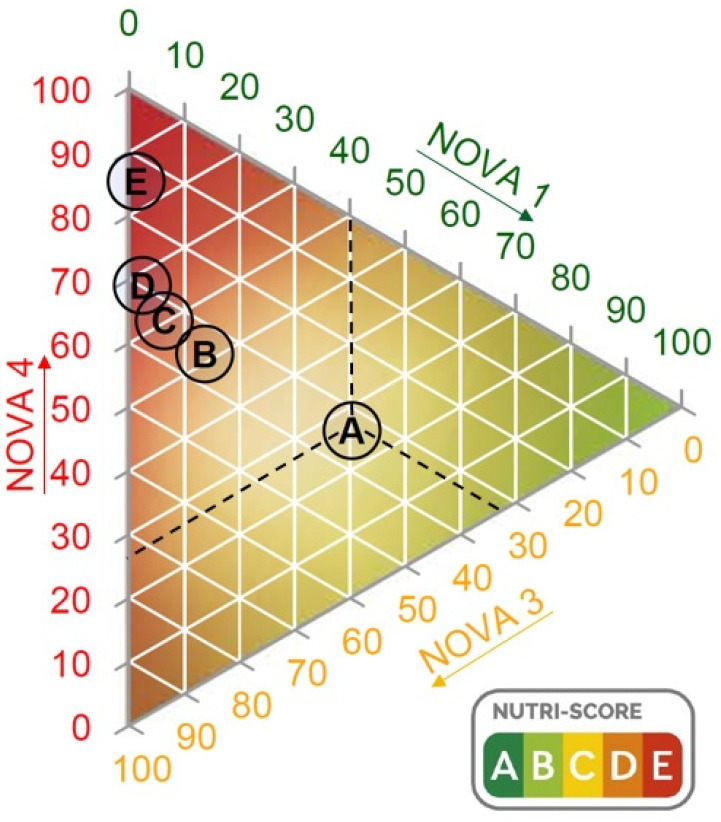
Ternary diagram representing the Nutri-Score categories in accordance with their positions on each of the three axes. Each axis represents the percentage of NOVA groups 1, 3 and 4. NOVA 2 is not shown as it only represented 1% of the data. The dashed lines indicate the coordinates of the different NOVA groups leading to the point where Nutri-Score category A is located (as an example for interpretation).

**Figure 4 nutrients-13-02783-f004:**
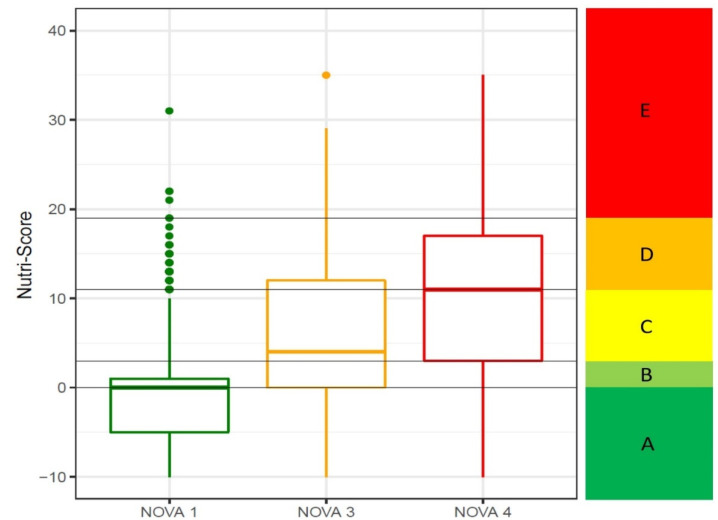
Boxplot showing the Nutri-Score values according to the different food processing categories (NOVA classification). NOVA 2 is not shown as it only represented 1% of the data. The box boundaries represent the interquartile range (IQR) and the line inside the box shows the median value. The black horizontal lines represent the score boundaries of each of the Nutri-Score categories.

**Table 1 nutrients-13-02783-t001:** Cross-frequency table between Nutri-Score and NOVA classification. Relative frequencies were calculated by rows.

Nutri-Score	NOVA 1 *	NOVA 2	NOVA 3 *	NOVA 4 *
**A**	846 (40.34%)	1 (0.06%)	703 (33.52%)	547 (26.08%)
**B**	218 (13.67%)	0 (0.00%)	556 (34.85%)	821 (51.48%)
**C**	142 (6.21%)	47 (2.07%)	746 (32.63%)	1351 (59.09%)
**D**	61 (2.34%)	18 (0.73%)	762 (29.54%)	1738 (67.39%)
**E**	17 (1.23%)	17 (1.26%)	190 (13.82%)	1150 (83.69%)

* These percentages have been represented in the ternary diagram (Figure 3).

**Table 2 nutrients-13-02783-t002:** Relative contribution of ultra-processed food groups (NOVA 4) in the Nutri-Score A classification.

Ultra-Processed Foods (NOVA 4) Classified as Nutri-Score A by Food Groups	*N* = 547
**Dairy products**	**21.92%**
Dairy desserts ^a^	31.19%
Flavoured yoghurts	28.44%
Skimmed and sweetened	40.37%
**Ready meals and canned dishes**	**18.72%**
**Vegetarian/vegan ready meals**	**14.63%**
**Flavoured plant-based drinks**	**10.68%**
**Pastries and cookies ^b^**	**8.06%**
**Ultra-processed breads**	**7.88%**
**Milkshakes and juice boxes**	**6.04%**
**0% jams and marmalades**	**4.85%**
**Protein products ^c^**	**3.12%**
**Sauces and dressings**	**2.70%**
**Ultra-processed cheese**	**1.40%**

^a^ Includes custard, flan or other fermented milk products; ^b^ Includes biscuits with sweeteners, wholemeal biscuits or cereal bars; ^c^ Includes protein shakes or yoghurts in different flavours.

## Data Availability

The data used in this study are available in Open Food Facts (https://world.openfoodfacts.org/, accessed on 24 February 2021), an open collaborative database of food products marketed worldwide, licensed under the Open Database License (ODBL).

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
