# Peer review of "Two Dimensions of Nutritional Value: Nutri-Score and NOVA"

_nutrients, 2021, doi:10.3390/nu13082783_

Round 1

Reviewer 1 Report

Two dimensions of nutritional value: Nutri-Score and NOVA

Carmen Romero Ferreiro et al.,

General comment

Based on the utility of front-of-pack labels to help consumers for choice of health foods, and as public health tool as well, the authors compare Nutriscore labelling with NOVA classification. The main finding is that all 5 categories of Nutriscore labelled foods contained also certain amount of foods ranked as ultraprocessed foods of NOVA 4, also known to have negative health impacts. They conclude on a necessary complement of Nutriscore labels indicating the level of processing applied to the food.

This paper is supposed to be included into a special issue “Human Health and Ultra-Processed Food” with Pr. M. Castellano as the Special issue Editor, perhaps targeted on “the role of their (ultraprocessed foods) marketing and advertisement”.

Main comments

1-      Even though recognizing the incompleteness of Nutriscore labels, the authors state that it could lead to misinterpretations of the consumers. This term “misinterpretation” does not reflect the data provided in the study. Indeed, more than a quarter of foods labelled as Nutriscore “A” are ultraprocessed but can be considered as healthy by the consumers. Within the Spanish market, it can concern 547 foods among 9,931 chosen for the study, and covering 11 food groups including plant-based ones usually considered by the consumers as healthy. The latter (vegetarian/vegan ready meals and flavored plant-based drinks, Table 2) represent almost 25% of the 547 foods selected as healthy with Nutriscore ranking. On the Table 2, “ultra-processed breads” is mentioned; whereas according to European definition, a bread contains only 3 ingredients, flour, water and baker’s yeast, it cannot be ultraprocessed. For these reasons, the term misinterpretations seem not be adapted to that situation. The authors should be clearer in expressing that Nutriscore includes unhealthy foods in their classification and, although their number is limited, this is more than just a misunderstanding for the consumer choosing these foods.

2-      After several papers published from 2017-2018, and several times confirmed since then in several countries including in the Spanish cohort SUN, this paper again recognizes that the level of processing is an important criteria for ranking foods form the less to the most healthier. It can be noticed that despite of these published data, European countries are still going on adopting Nutriscore as front-of-pack labels.

Going on processing versus nutrient composition, an interesting data is shown in Figure 4, but not enough explored and discussed. It shows that for Nova 4 foods, a median value of 11 is obtained on Nutriscore scale ranging from -10 to 40.

-          A first interpretation can be to consider that the foods ranked as “D” and “E”, and Nova 4 should be avoided from the diet, altogether they represent 2,848 items out of 9,931 chosen in the Spanish market. Together with the 547 items, already mentioned above, one can consider that 3,395 food items are sold (usually with aggressive advertisements) as unhealthy foods, representing almost one third of all 9,931 foods. The present paper should analyze quantitatively the level of unhealthy foods in Spain in order to give an explanation of a sentence in the introduction section (lines 31-33) “Spain, despite following a Mediterranean diet with beneficial health effects, has to deal with nutritional problems that have a substantial human, social and economic cost”. In that sentence, the authors just forgot that Spain, as all European countries, made a nutrition transition from local and traditional to industrial and globalized foods (even imitating Spanish meals).

-          A second interpretation is that processing clearly alter the functionality of the food leading them to be unhealthy despite of satisfying nutritional property estimated from their nutrient content. The weight of processing seems even heavier that nutrient contents. There is a meaning under this data: enteral and parenteral nutrient solution, even if they are balanced with all known nutrients, cannot compensate the physiological effect of eating foods.

3-      About the conclusion (lines 245-246) that Nutriscore “should be complemented by labels indicating the degree of food processing to enable consumers to make better choices”, the authors did not mention whether this question has been already investigated and published.  It seems that papers proposed “Siga classification” as a solution on science-evidence based analysis. This work should be mentioned, with its strength and weakness, instead of letting believing the readers that such a way has never been explored.

Conclusion:

This paper need deep revision and is not suitable for a publication in its present form. It starts from the issue of the growth of the most prevalent chronic non-communicable diseases (in Spain) to conclude on complementing the Nutriscore ranking of the foods. It is not sufficient and not satisfactory from the nice work done and the data obtained.

The data may help to quantitatively estimate the level of unhealthy foods available in Spanish market. This point has been neglected. The following sentence « Consequently, the Nutri-Score system, as seen in some studies [26], offers the potential to boost healthy products (Nutri-Score A), without affecting the consumption of unhealthy products as it includes a high percentage of them in all the remaining categories (Nutri-Score B, C, D and E)” indirectly support Nutriscore and is out of the scope of this study. Formally and based on the findings of the present study, the only way to “boost” is to include in a diet 1,550 (Nutriscore “A” minus Nova 4) out of 9,931 foods proposed in the market, representing only 15.6% of the offer.

The authors refer to health for Nova but not for Nutriscore. A sentence on Nutriscore-health relationship should be addressed in order to compare both classifications towards health criteria. In other word, foods ranked in “A” are not clearly stated as healthy or unhealthy, even if the ranking and the color given to those products let suppose that they are healthy. In that case, the study made by the authors should mention that the product ranked “A” by Nutrisore contained a limited but significant number of unhealthy foods that include also plant-based ones usually considered as healthy by the consumers.

The data also show that food processing and nutritional quality are not only complementary dimensions; the present study should estimate and discuss the weight of processing over nutrient from the data shown in Figure 4.

Finally yet importantly, they open research question to complement Nutriscore with processing whereas this question has been answered in published papers (see Siga classification as key word).

With the above remarks, this paper can bring a real support for the special issue “Human Health and Ultra-Processed Food”, opening the question of “the role of their marketing and advertisement”, largely applied by the industries of ultraprocessed foods. One can attenuate this commercial approach with front-of-pack labels, other than the ones that include ultra-processed foods in their ranking.

Author Response

Response to Reviewer 1 Comments

We are very thankful for the comments and suggestions. We think that we have made all the changes proposed and we hope the manuscript has improved notably now. Below each concern you can find the answer proposed and the section and paragraph where we have made the changes. Besides, we have made a style and grammar check all along the text.

General comment

Based on the utility of front-of-pack labels to help consumers for choice of health foods, and as public health tool as well, the authors compare Nutriscore labelling with NOVA classification. The main finding is that all 5 categories of Nutriscore labelled foods contained also certain amount of foods ranked as ultraprocessed foods of NOVA 4, also known to have negative health impacts. They conclude on a necessary complement of Nutriscore labels indicating the level of processing applied to the food.

This paper is supposed to be included into a special issue “Human Health and Ultra-Processed Food” with Pr. M. Castellano as the Special issue Editor, perhaps targeted on “the role of their (ultraprocessed foods) marketing and advertisement”.

Main comments

1-      Even though recognizing the incompleteness of Nutriscore labels, the authors state that it could lead to misinterpretations of the consumers. This term “misinterpretation” does not reflect the data provided in the study. Indeed, more than a quarter of foods labelled as Nutriscore “A” are ultraprocessed but can be considered as healthy by the consumers. Within the Spanish market, it can concern 547 foods among 9,931 chosen for the study, and covering 11 food groups including plant-based ones usually considered by the consumers as healthy. The latter (vegetarian/vegan ready meals and flavored plant-based drinks, Table 2) represent almost 25% of the 547 foods selected as healthy with Nutriscore ranking. On the Table 2, “ultra-processed breads” is mentioned; whereas according to European definition, a bread contains only 3 ingredients, flour, water and baker’s yeast, it cannot be ultraprocessed. For these reasons, the term misinterpretations seem not be adapted to that situation. The authors should be clearer in expressing that Nutriscore includes unhealthy foods in their classification and, although their number is limited, this is more than just a misunderstanding for the consumer choosing these foods.

We completely agree with this comment. We have changed the term “misinterpretation” in the manuscript and explained that the Nutri-Score system does not identify all unhealthy foods (as it includes them in all its categories) that may be nutritionally well-rated, even if they are highly processed and contain many additives. We have clarified this in the discussion (line 215-216 and line 229-230) and in the conclusions (line 291). In addition, we have specified that category A includes a significant amount of unhealthy products (line 246-247).

2-      After several papers published from 2017-2018, and several times confirmed since then in several countries including in the Spanish cohort SUN, this paper again recognizes that the level of processing is an important criteria for ranking foods form the less to the most healthier. It can be noticed that despite of these published data, European countries are still going on adopting Nutriscore as front-of-pack labels.

Going on processing versus nutrient composition, an interesting data is shown in Figure 4, but not enough explored and discussed. It shows that for Nova 4 foods, a median value of 11 is obtained on Nutriscore scale ranging from -10 to 40.

-          A first interpretation can be to consider that the foods ranked as “D” and “E”, and Nova 4 should be avoided from the diet, altogether they represent 2,848 items out of 9,931 chosen in the Spanish market. Together with the 547 items, already mentioned above, one can consider that 3,395 food items are sold (usually with aggressive advertisements) as unhealthy foods, representing almost one third of all 9,931 foods. The present paper should analyze quantitatively the level of unhealthy foods in Spain in order to give an explanation of a sentence in the introduction section (lines 31-33) “Spain, despite following a Mediterranean diet with beneficial health effects, has to deal with nutritional problems that have a substantial human, social and economic cost”. In that sentence, the authors just forgot that Spain, as all European countries, made a nutrition transition from local and traditional to industrial and globalized foods (even imitating Spanish meals).

-          A second interpretation is that processing clearly alter the functionality of the food leading them to be unhealthy despite of satisfying nutritional property estimated from their nutrient content. The weight of processing seems even heavier that nutrient contents. There is a meaning under this data: enteral and parenteral nutrient solution, even if they are balanced with all known nutrients, cannot compensate the physiological effect of eating foods.

Certainly, we should have included a more detailed explanation of the Figure 4. We have now mentioned these findings in the discussion. We have quantified the level of unhealthy foods in Spain, and have provided an interpretation of why, despite following a Mediterranean diet in Spain, we encounter nutritional health problems (line 262-271). We have also discussed the findings that ultra-processed foods scored so low on the Nutri-Score (line 252-261).

3-      About the conclusion (lines 245-246) that Nutriscore “should be complemented by labels indicating the degree of food processing to enable consumers to make better choices”, the authors did not mention whether this question has been already investigated and published.  It seems that papers proposed “Siga classification” as a solution on science-evidence based analysis. This work should be mentioned, with its strength and weakness, instead of letting believing the readers that such a way has never been explored.

We thank reviewer #2 for this suggestion. It has been included in the revised manuscript the existence of some tools that could provide a solution to this problem, such as the SIGA classification, and added the relevant references (discussion line 275-278; conclusions line 292-293).

Conclusion:

This paper need deep revision and is not suitable for a publication in its present form. It starts from the issue of the growth of the most prevalent chronic non-communicable diseases (in Spain) to conclude on complementing the Nutriscore ranking of the foods. It is not sufficient and not satisfactory from the nice work done and the data obtained.

The data may help to quantitatively estimate the level of unhealthy foods available in Spanish market. This point has been neglected. The following sentence « Consequently, the Nutri-Score system, as seen in some studies [26], offers the potential to boost healthy products (Nutri-Score A), without affecting the consumption of unhealthy products as it includes a high percentage of them in all the remaining categories (Nutri-Score B, C, D and E)” indirectly support Nutriscore and is out of the scope of this study. Formally and based on the findings of the present study, the only way to “boost” is to include in a diet 1,550 (Nutriscore “A” minus Nova 4) out of 9,931 foods proposed in the market, representing only 15.6% of the offer.

We thank the reviewer for this highly pertinent remark. It is true that the sentence is out of the scope of this study. We have now removed that sentence.

The authors refer to health for Nova but not for Nutriscore. A sentence on Nutriscore-health relationship should be addressed in order to compare both classifications towards health criteria. In other word, foods ranked in “A” are not clearly stated as healthy or unhealthy, even if the ranking and the color given to those products let suppose that they are healthy. In that case, the study made by the authors should mention that the product ranked “A” by Nutrisore contained a limited but significant number of unhealthy foods that include also plant-based ones usually considered as healthy by the consumers.

We appreciate this suggestion. A sentence relating the Nutri-Score to health has been included in the introduction (line 53-56). In addition, we have specified that category A includes a significant amount of unhealthy products (line 246-247), and we have highlighted plant-based products (line 249-251). We have also included an example of a prepared legume dish in an attempt to also cover the category of plant-based meals (line 242-245).

The data also show that food processing and nutritional quality are not only complementary dimensions; the present study should estimate and discuss the weight of processing over nutrient from the data shown in Figure 4.

Following this suggestion, we have discussed the weight of processing over nutrient (line 252-261).

Finally yet importantly, they open research question to complement Nutriscore with processing whereas this question has been answered in published papers (see Siga classification as key word).

We have included the SIGA classification in the new version of the manuscript (discussion line 275-278; conclusions line 292-293).

With the above remarks, this paper can bring a real support for the special issue “Human Health and Ultra-Processed Food”, opening the question of “the role of their marketing and advertisement”, largely applied by the industries of ultraprocessed foods. One can attenuate this commercial approach with front-of-pack labels, other than the ones that include ultra-processed foods in their ranking.

Reviewer 2 Report

1) Title: capitalize the first letter of every word, use a recently published paper as a template;

2) Verity the format of affiliations and correspondence, use a recently published paper as a template;

3) Line 11: consumers instead of shoppers;

4) Line 20: have instead of has;

5) Keywords: Ultra-Processed Foods instead of ”Ultra-processed foods”;

6) Keywords: There is no need to add Nutri-Score, NOVA in this list as they are in the title. They may be replaced by other words;

7) Line 29: More recent references should be added. For example, FAO, IFAD, UNICEF, WFP and WHO. 2021. The State of Food Security and Nutrition in the World 2021. Transforming food systems for food security, improved nutrition and affordable healthy diets for all. Rome, FAO. https://doi.org/10.4060/cb4474en

8) Line 30-33: More recent references should be added;

9) Line 44: reformulate the sentence; The reason why new labels are already being implemented...;

10) Line 175-182: Better interpretation could be given to figure 4;

11) Line 194: Please replace “other study where they” by “another study where the authors”;

12) Line 217-224: only one example (on dairy products) is discussed, while the authors are highly encouraged to discuss at least an example of each food category (meat and seafood products, eggs, seeds, etc.)

Author Response

Response to Reviewer 2 Comments

We are very thankful for the comments and suggestions. We think that we have made all the changes proposed and we hope the manuscript has improved notably now. Below each concern you can find the answer proposed and the section and paragraph where we have made the changes. Besides, we have made a style and grammar check all along the text.

1) Title: capitalize the first letter of every word, use a recently published paper as a template;

Following this suggestion, we have modified the title.

2) Verity the format of affiliations and correspondence, use a recently published paper as a template;

We thank reviewer #2 for this remark. We have checked the format of the affiliations and modified it.

3) Line 11: consumers instead of shoppers;

We have modified this information in the text (line 13)

4) Line 20: have instead of has;

We have modified this information in the text (line 23)

5) Keywords: Ultra-Processed Foods instead of ”Ultra-processed foods”;

We have modified this keyword (line 26).

6) Keywords: There is no need to add Nutri-Score, NOVA in this list as they are in the title. They may be replaced by other words;

Following this suggestion, we have updated the keywords. We have removed Nutri-Score and NOVA and added Food Quality.

7) Line 29: More recent references should be added. For example, FAO, IFAD, UNICEF, WFP and WHO. 2021. The State of Food Security and Nutrition in the World 2021. Transforming food systems for food security, improved nutrition and affordable healthy diets for all. Rome, FAO. https://doi.org/10.4060/cb4474en

We thank reviewer #2 the suggestion of this new reference that has been published recently. We have included the above reference and another one (line 33-34, references added: 2 and 3).

  1. The State of Food Security and Nutrition in the World 2021; FAO, IFAD, UNICEF, WFP and WHO, 2021; ISBN 978-92-5-134325-8.
  2. United Nations Political Declaration of the High-Level Meeting of the General Assembly on the Prevention and Control of Non-Communicable Diseases 2011.

8) Line 30-33: More recent references should be added;

According to reviewer #2, we have included recent references that support these claims (line 35-36, references added: 5, 6 and 8).

  1. Stuckler, D.; McKee, M.; Ebrahim, S.; Basu, S. Manufacturing Epidemics: The Role of Global Producers in Increased Consumption of Unhealthy Commodities Including Processed Foods, Alcohol, and Tobacco. PLoS Med 2012, 9, e1001235, doi:10.1371/journal.pmed.1001235.
  2. Cena, H.; Calder, P.C. Defining a Healthy Diet: Evidence for the Role of Contemporary Dietary Patterns in Health and Disease. Nutrients 2020, 12, 334, doi:10.3390/nu12020334.

  1. Sofi, F.; Macchi, C.; Abbate, R.; Gensini, G.F.; Casini, A. Mediterranean Diet and Health Status: An Updated Meta-Analysis and a Proposal for a Literature-Based Adherence Score. Public Health Nutr 2014, 17, 2769–2782, doi:10.1017/S1368980013003169.

9) Line 44: reformulate the sentence; The reason why new labels are already being implemented...;

Following this suggestion, we have reformulated this sentence (line 49-50).

10) Line 175-182: Better interpretation could be given to figure 4;

We agree with this comment. Figure 4 is not sufficiently explored or discussed. We have added in the discussion a possible interpretation of this figure, as food processing affects the functionality of food by making it unhealthy despite satisfying the nutritional properties estimated from its nutrient content. This is why ultra-processed foods (NOVA 4) score so low in the Nutri-Score which focuses mainly on nutrient composition. (line 252-261)

11) Line 194: Please replace “other study where they” by “another study where the authors”;

This sentence has been replaced in the new version of the manuscript (line 208)

12) Line 217-224: only one example (on dairy products) is discussed, while the authors are highly encouraged to discuss at least an example of each food category (meat and seafood products, eggs, seeds, etc.)

We thank reviewer #2 for this suggestion. We have tried to include one more example since it is difficult to discuss examples of foods in a generic way (without naming any specific brand or product). We have included an example of ready meals, as it is the second most frequent ultra-processed food group that is classified as Nutri-Score A, the example included is ready meals of pulses. We have selected this example to try to cover also the category of plant-based meals, which is the third most frequently misclassified group (Table 2). (line 242-245)

Round 2

Reviewer 1 Report

Thank you for this revision that alo improves the paper.

Author Response

We are very thankful for the time and effort taken and the suggestions made.